# The Impact of Slumping Oil Price on the Situation of Tanker Shipping along the Maritime Silk Road

**Naixia Mou** [1,2], **Yanxin Xie** [1], **Tengfei Yang** [3], **Hengcai Zhang** [2,*] **and Yoo Ri Kim** [4]

1   College of Geomatics, Shandong University of Science and Technology, Qingdao 266590, China
2   State Key Laboratory of Resources and Environmental Information System, Institute of Geographical Sciences and Natural Resources Research, Chinese Academy of Sciences, Beijing 100101, China
3   Institute of Remote Sensing and Digital Earth, Chinese Academy of Sciences, Beijing 100094, China
4   School of Hospitality and Tourism Management, Faculty of Arts and Social Sciences, University of Surrey, Guildford GU2 7UU, UK
*   Correspondence: zhanghc@lreis.ac.cn

**Abstract:** Nearly 70% of the world's maritime crude oil transportation relies on the Maritime Silk Road (MSR). In order to deeply explore the impact of slumping oil price on the shipping situation of tanker along the MSR, this paper establishes the relationship between monthly ship and oil price through Autoregressive Distributed Lag model. Distributions of cargo flow before and after the oil price slumped are compared to explore the changing law of tanker shipping situation. The study finds: (1) The correlation between the cargo flow situation of the tanker seaborne export and oil price, where the export cargo flow correlation is stronger than that of the import cargo flow. (2) The MSR tanker shipping situation is lagging (3 months) behind the impact of oil price. The lag effect in Europe, North Asia and East Asia is strong while that in Southeast Asia and South Asia is weak. (3) After the oil price slumped, the tanker shipping cargo flow increased less during the crude oil export stage, and the increase in the crude oil shipping trade after the transfer period was larger. The research results can provide a scientific basis for improving the decision-making ability of the crude oil shipping market and formulating maritime operations management measures.

**Keywords:** Maritime Silk Road; oil price; tanker shipping situation; ARDL model; cargo flow

## 1. Introduction

The 21st Century Maritime Silk Road (MSR), originated from the ancient Maritime Silk Road, is an important channel to promote the smooth trade between Asia, Europe and Africa. It is an emerging trade route connecting China to the world under the changing situations of global politics and trade patterns. At present, the coverage of MSR continues to expand, the maritime trade along its route accounts for more than 35% of the global merchandise trade about 70% of the world's maritime crude oil transport [1].

Petroleum energy is an important fossil energy, and its unbalanced distribution and supporting role in economic development determines its importance in the world economy and trade [2]. As an important factor affecting global economic activities, oil price has an obvious impact on the maritime trade, especially on the maritime crude oil trade. Oil price fluctuations directly stimulate oil demand [3], and then affect the change of tanker shipping situation. In particular, the global oil consumption powers, such as the United States [4], China [5], Japan [6], South Korea [7], etc., have more significant changes in strategic oil reserves and consumption. At the same time, shipping power relies heavily on fossil fuel, making freight rates and transport costs more vulnerable to oil price shocks [8,9]. For the shipping industry, fuel costs account for a large proportion of ship operating costs. Therefore, it is of practical

significance to rationally adjust fuel management policies to reduce the ship operating costs, enhance the interests of shipping companies and promote the sustainable development of shipping industry [10].

Current researches focus on the long-term relationship between oil price fluctuation and maritime trade [11–13], while those have less concern about the short-term impact of oil price surge and slump on maritime transport. Therefore, taking the oil price slump in 2014 as an example, this paper attempts to supplement the deficiency of the related research by studying the short-term change relationship between the oil price slump and the shipping situation of oil tankers. This paper studies the dynamic impact of slumping oil price on the shipping situation of tanker along the MSR by comparing the impact differences of different regions, analyzes the changes of tanker cargo flows before and after the oil price slumped at the port scale from the angle of crude oil import, export and transshipment. This reveals the development rule of tanker shipping situation in the short-term and puts forward reasonable suggestions for the optimization of MSR tanker shipping network, port operations and the sustainable development of maritime crude oil trade.

The rest of this paper is organized as follows: Section 2 includes literature review and contribution summary; Section 3 mainly introduces study areas, data sources and methods; Section 4 presents empirical results and related analysis; followed by discussion of empirical results in Section 5; and lastly, Section 6 concludes the paper.

## 2. Literature Review

With the rapid development of social economy, the relationship between oil price and global economic activities becomes closer. A series of studies have aimed at exploring the potential relationship between oil price and macroeconomic variables, such as GDP [14], stock market [15], exchange rate [16], interest rate [17], commodity market [18], agricultural and metal commodity price [19], food price [20], gasoline, and natural gas price [21,22]. Research on oil price and shipping mainly include crude oil production [23], tanker market variable [24], Baltic Dirty Tanker Index (BDTI) [23], Baltic Dry Index (BDI) [25] and so on. Container ships, tankers and dry bulk carriers are the main types of shipping trade. Container transport supports the global trade supply chain as an important way of global port trade [26,27]. Oil tanker transportation is mainly used for maritime trade in oil and refined oil [28,29]. Dry bulk transport serves global maritime trade in bulk commodities such as iron ore, coal, food and so on [30,31]. The dynamic changes of three types of cargo ships can effectively reflect the characteristics of the changes of maritime trade among ports, countries and regions. Therefore, this paper chooses the trajectory of oil tankers to explore the relationship between shipping and oil price.

In related research of maritime transportation, scholars mainly use data mining methods to reveal the characteristics of the maritime network structure [32], the maritime network evolution mechanism [33], the change law and the evolution mode of the traffic flow in the sea [34] through constructing maritime network, so as to master the changes of the pattern of maritime trade. Mou et al. [35] have revealed the relevance and cargo flow mode of regional maritime trade patterns along the MSR through different types of maritime networks. Peng et al. [36] discussed the change of the spatio-temporal characteristics of the global maritime crude oil trade with the "Hub-and-Spoke" structure of the tanker maritime trade. Hao et al. [37] constructed a global network of fossil energy flows with countries as nodes and found that trade relations between specific countries played a key role in fossil energy trade. At the same time, they formed a major fossil energy trade group with the Russian Federation, the United States, Japan and Saudi Arabia as the core. By means of the complex network index and traffic flow, these studies reflect the model of maritime trade evolution and the distribution of the patterns and changes in energy flow. Based on the actual trajectory of the tanker, this paper attempts to explore the change law of the impact of the oil price slumped from the perspective of the change of crude oil cargo flow.

At present, most studies mainly use data such as port operation record [38], freight cost [39], maritime transport index, etc., to establish the relationship between maritime trade and oil price, which verified that the oil tanker shipping market is more significantly affected by oil price fluctuations.

Chen et al. [40] used Multifractal Detrended Cross-correlation Analysis (MF-DCCA) to study the correlation between West Texas International Crude Oil Price (WTI) and BDTI, which showed that the short-term cross-correlation is higher than the long-term correlation. Chen et al. [39] analyzed the changes of crude oil imports, BDI and the proportion of fuel cost in transportation cost, which revealed the inherent relationship between oil price fluctuation and Chinese shipping industry. These studies are mainly based on macro-shipping transport statistics. By contrast, the trajectory data of oil tankers selected in this paper can truly reflect the change of tanker situation and explore the impact of slumping oil price on the tanker maritime network and cargo flows from the perspective of import and export cargo flows. As an important global economic activity, crude oil trade has gradually evolved into a long-term stable and orderly system, as well as a long-term stable relationship between oil price and tanker transportation. Some scholars regard oil price and maritime trade as two long-time series variables and use lag models to study the long-term relationship between them. Shi et al. [23] divided crude oil price shocks into crude oil supply shocks and non-supply shocks (transportation costs). The structural vector autoregressive (SVAR) model and impulse response were used to analyze the response of oil tanker market to different shocks. The results showed that the impact of crude oil supply shocks on the oil tanker market in the same period is significant, while the impact of non-supply shock is weak. Altinay [41] estimated the long-term and short-term changes of the Turkish crude oil demand by using the autoregressive distribution lag boundary test method. The results showed that its crude oil demand is less affected by the fluctuation of oil price. Other scholars take oil price as an external factor to test whether there is a bidirectional relationship between oil price fluctuation and the structure of maritime network and traffic flow. Findings have shown that oil price fluctuation causes changes in the structure of tanker maritime network and traffic flow [42]. These studies focus on individual countries, lacking the whole variation trend of the maritime crude oil transport network system and the comparison of differences in import and export trade between different regions and ports. Moreover, long-term fluctuations tend to overlook the short-term impact of oil price surge and slump on the oil tanker shipping industry, in which this paper aims to examine.

Previous studies have often explored the long-term relationship between oil price fluctuation and maritime trade through different index variables and research methods, but there have been little interests on the impact of oil price surge and slump on the situation of oil tanker shipping in the short-term; this impact on crude oil import, export and transshipment is also rarely examined. Therefore, this paper focuses on a time period before and after the oil price slumped in 2014. Based on Automatic Identification System (AIS) tanker trajectory data and monthly variation as the time scale, the dynamic impact of slumping oil price on the tanker shipping situation along the MSR was studied by using the Spearman rank correlation method and autoregressive distribution lag model and by comparing the impact differences of different regions. At the same time, based on the results of the lag period, we analyzed the changes of tanker cargo flow before and after the oil price slumped at the port scale from the angle of crude oil import, export and transshipment. The research results can provide references for comprehensively understanding the sustainable development trend of marine trade along the MSR, formulating strategic measures for oil tanker transportation by countries importing and exporting crude oil, formulating management measures for maritime transportation by oil transport companies and operating ports, and avoiding market risks.

## 3. Data and Methods

### 3.1. Study Areas and Data

The existing literature divides the world into 13 regions based on cultural, religious, linguistic, geographical and other factors, including East Asia, Southeast Asia, Central Asia, West Asia, North Africa, Sub-Saharan Africa, Eastern and Western Europe, and North Asia. North America, Latin America, Oceania, Antarctica. There are eight research areas along the MSR [43], with a total of 1879 ports, as is shown in Figure 1.

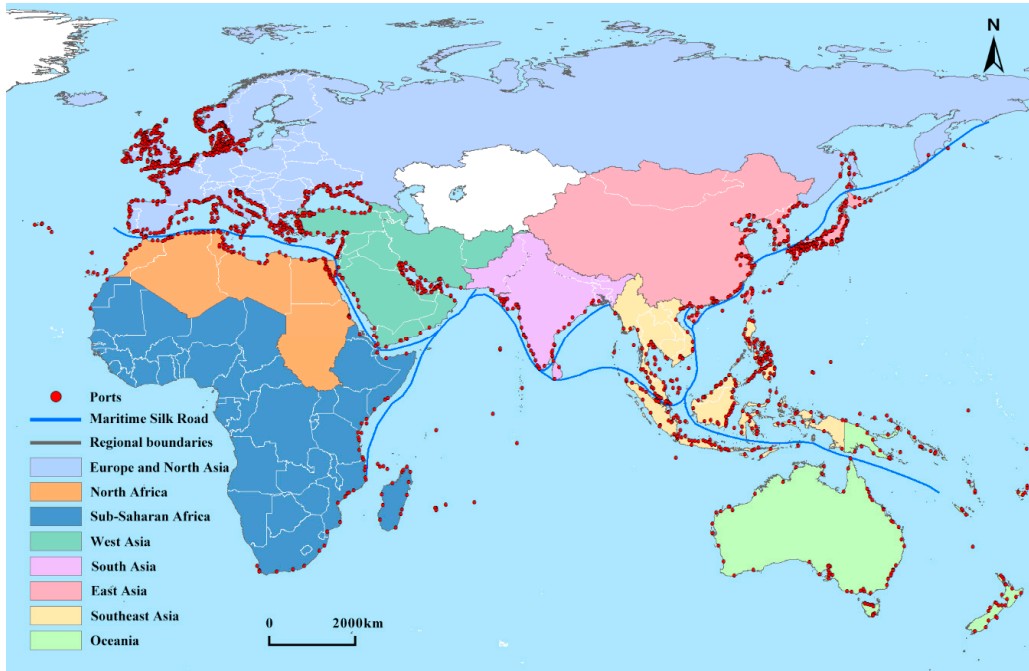

**Figure 1.** The study areas.

The oil price shock in the second half of 2014 was the most significant macroeconomic shock in recent years. To explore the changes of crude oil shipping trade before and after the oil price slumped, we extract the records of arrival and departure of ships based on the AIS data of tankers from 1 January 2014 to 31 March 2015. By combining with the global port index data published by National Geospatial-Intelligence Agency and taking each arrival and departure of each ship as an origin–destination (OD) data, we get 427,267 OD data of tankers. As Figure 2 shows, OD data of tankers intuitively reflect the distribution of crude oil trade between ports along the MSR. Additionally, the maritime trade in East Asia, the Strait of Malacca, the Persian Gulf and the Mediterranean Sea are relatively intensive. Based on OD data of tankers, taking the arrival and departures of a port as the ship frequency in the port, and then superimposing the ship frequency in ports belonging to the same region, the monthly number of tankers in each region and the whole along MSR is finally obtained, which can be used to quantitatively express the frequency change of tanker shipping situation in the study region. The monthly oil price data is derived from the U.S. Energy Information Administration (EIA). The monthly oil price and the MSR overall tanker change during the study period are shown in Figure 3.

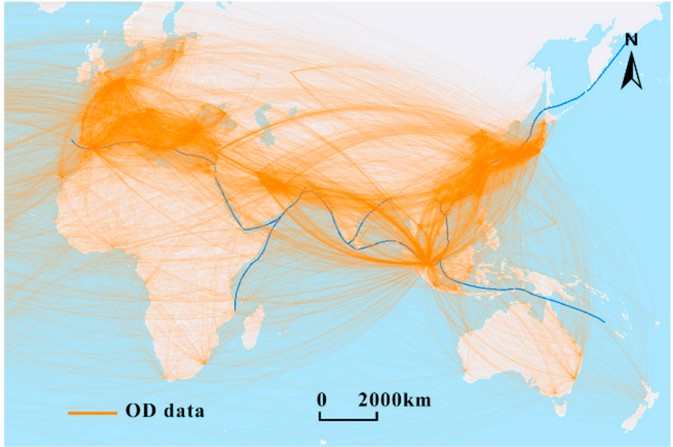

**Figure 2.** Tanker OD trajectory.

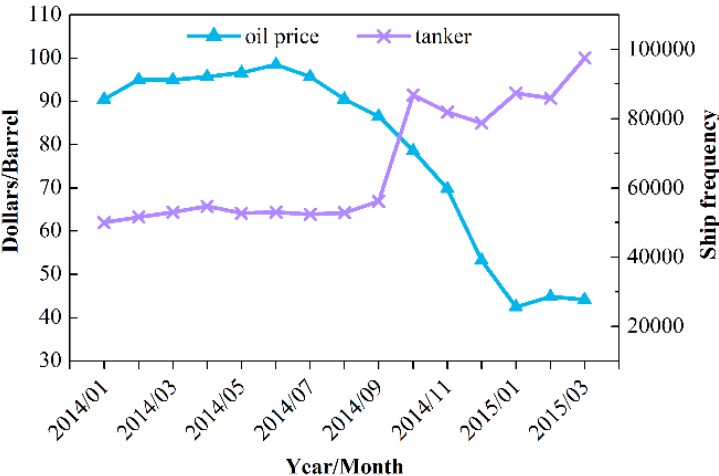

**Figure 3.** Changes of oil price and the ship frequencies.

### 3.2. Methods

#### 3.2.1. Spearman Rank Correlation Analysis

Spearman rank correlation analysis is a nonparametric statistical correlation analysis method, which uses Spearman rank correlation coefficient $r_s$ to measure the rank correlation strength between variables. It is often used to measure the non-linear monotonic relationship [44]. In order to analyze the correlation between ship frequency and oil price in different regions, this paper uses Spearman rank correlation to carry out the analysis test after judging that two variables have a non-linear relationship (which does not satisfy the hypothetical condition of Pearson), and also due to the small sample size. In practical calculation, the difference of rank is used to calculate the value of $r_s$. Assuming that the original data $x_i$ and $y_i$ have been arranged in descending order and that $x'_i$ and $y'_i$ are the positions of the original data $x_i$ and $y_i$ after the arrangement, $x'_i$ and $y'_i$ are called the ranks of the variables $x_i$ and $y_i$, and $d_i = x'_i - y'_i$ is the difference of ranks of $x_i$ and $y_i$. In the case of different ranks before and after sorting, $r_s$ can be calculated by the Equation (1):

$$r_s = 1 - \frac{6\sum_{i=1}^{n} d_i^2}{n(n^2 - 1)} \tag{1}$$

In the case of the same rank before and after ordering, the linear correlation coefficients of Pearson between ranks should be calculated according to Equation (2), where $\overline{x}$ and $\overline{y}$ is the mean of the sample variables.

$$r_s = \frac{\sum_{i=1}^{n} (x_i - \overline{x})(y_i - \overline{y})}{\sqrt{\sum_{i=1}^{n} (x_i - \overline{x})^2 \sum_{i=1}^{n} (y_i - \overline{y})^2}} \tag{2}$$

#### 3.2.2. ARDL Lag Model

Autoregression is a regression of a variable $Q_t$ with its own lag term, referred to as the Autoregressive Model. When a variable is estimated by the AR model and is also affected by the current and lag values of other variables $P_t$, the model is the autoregressive distribution lag (ARDL) model for the event analysis of hysteresis effects [45–47]. This paper uses a traditional linear ARDL model constructed by Pesaran et al., which is suitable for the small sample data in this paper [48]. In order to reduce the volatility of the original data and the difference between the statistical data and avoid heteroscedasticity, we have carried out logarithmic transformations of the oil price and the ship

frequency variables, which do not affect the cointegration relationship of the data fitting. The equation expression for the ARDL model of oil price and ship frequency is:

$$\ln Q_t = c + \sum_{i=1}^{k} \alpha_i \ln Q_{t-i} + \sum_{i=1}^{k} \beta_i \ln P_{t-i} + \varepsilon_t \tag{3}$$

Ordinary Least Squares (OLS) Euler algorithm is used to estimate the parameters of the model. At the same time, the fitting effect and lag length $k$ of the model are determined according to Akaike Information Criterion (AIC) and Schwarz Criterion (SC). Where $Q_t$ is the statistical value of the $t$-month tanker ship, $\ln Q_t$ is the natural logarithm of the ship frequency variable of $t$-month, that is, the dependent variable; $\ln Q_{t-i}$ and $\ln P_{t-i}$ are the ship frequency variables of the $i$-order lag, and the natural logarithm of the oil price variable of the $i$-order lag, that are, the explanatory variables; $\alpha_i$ and $\beta_i$ are the lag model coefficients of ship frequency and the oil price, respectively; $k$ is the maximum lag order of the model; $c$ is a constant term; $\varepsilon_t$ is a random term.

## 4. Empirical Results

### 4.1. Nonlinear Correlation

As is shown in Figure 4, based on the monthly variations of the overall tanker flow along the MSR and the distribution of oil price during the same period, and (a), (b) and (c) correspond to the import and export cargo flows, import cargo flows and export cargo flows, respectively. It can be seen from Figure 3 that the variables of ship frequency and oil price are not continuous. We can see from the Figure 4 that the relationship between oil price and ship frequency is non-linear and does not satisfy the hypothetical condition of Pearson. For the above reasons, in this paper, the Spearman rank correlation analysis method is suitable to be used to calculate the correlation between ship frequency and the oil price.

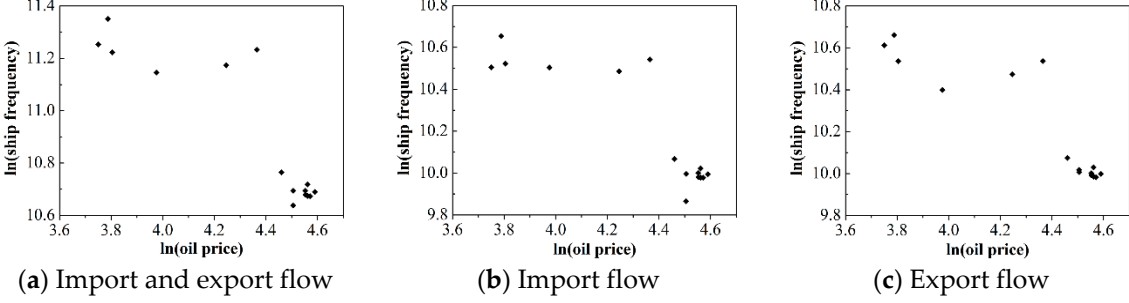

(**a**) Import and export flow          (**b**) Import flow          (**c**) Export flow

**Figure 4.** Oil price and cargo flow distribution.

The results of the Spearman rank correlation analysis are shown in Table 1. There is a significant negative correlation between the total import and export volume of the tanker, the import cargo flow and the export cargo flow along the MSR, that is, when the oil price slumped, the frequency of tanker ships increases, and the shipping situation rose accordingly. At the same time, the negative correlation between oil price and export cargo flow situation is stronger in magnitude than that of import and export, and the correlation between oil price and import cargo flow situation is slightly weaker than that of import and export.

In the areas along the MSR, the absolute value of the correlation coefficient between ship frequency and oil price in most regions is greater than 0.5, which indicates that the oil price slump has a significant impact on the shipping situation along the MSR. As is shown in Figure 5, the degree of correlation shows a "weak-strong-weak" distribution from the north to the south, and a "weak-medium-strong" class distribution from the west to the east. Among the eight regions, the important transshipment area for maritime crude oil trade, ship frequency in Southeast Asia has the strongest negative correlation

with oil price, with a correlation coefficient of −0.871; ship frequency in North Africa has the weakest negative correlation with oil price, with a correlation coefficient of −0.432. The correlation of the Asian region (East Asia, Southeast Asia, South Asia) is greater, with the correlation of Europe and North Asia, sub-Saharan Africa are second. As well, West Asia, North Africa and Oceania have a weaker correlation.

**Table 1.** Correlation calculations of tanker flow.

| Type | Correlation Coefficient $r_s$ | Significance Test $P_s$ |
|---|---|---|
| Import and export flow | −0.807143 ** | 0.000275 |
| Import flow | −0.782143 ** | 0.00057 |
| Export flow | −0.889286 ** | 0.000009 |

Note: ** indicates significant at 1% confidence level.

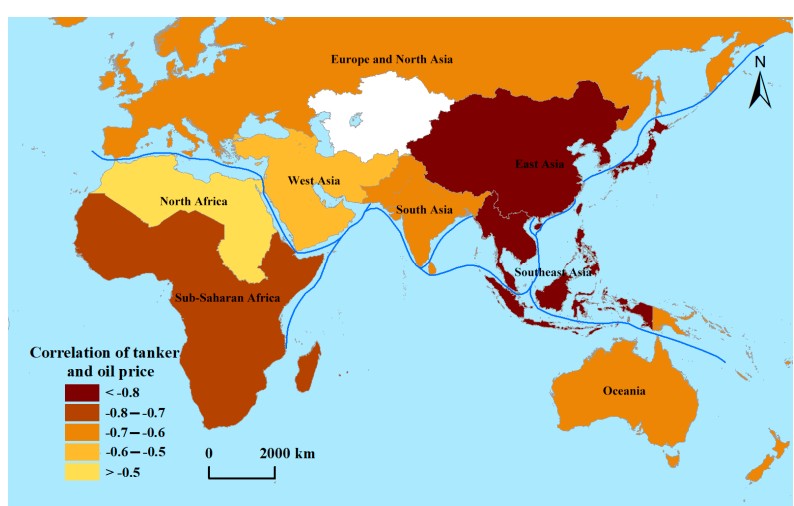

**Figure 5.** Correlation of study areas.

As far as the import and export cargo flow of oil tankers are concerned, the negative correlation between ship frequency change of export cargo flow and oil price along the MSR is generally stronger than that of import cargo flow. As is shown in Figure 6, the regions with a strong negative correlation of tanker import cargo flow are mainly concentrated in Southeast Asia, East Asia and sub-Saharan Africa; the regions with a strong negative correlation of oil tanker export cargo flow mainly include Southeast Asia, East Asia, Europe and North Asia. Besides, countries with large demand for petroleum resources are mainly distributed in East Asia and Europe. The frequency of maritime crude oil trade in the ports of these countries is higher. In addition to trade with crude oil export ports and transit ports, crude oil transport between countries and within countries is more frequent. Ports in Southeast Asia are important hubs linking East Asia with West Asia, Europe and Africa. In addition to meeting their own demand for crude oil imports, they also undertake the transshipment function of the current maritime crude oil trade. Therefore, the slumping oil price has a more obvious impact on the tanker shipping situation in these areas. Comparing the correlation of import and export cargo flow, we can see that the degree of correlation for both import and export in most regions is consistent: for example, the correlation between tanker number and oil price in Southeast Asia is the strongest, the correlation coefficients are −0.846 and −0.896, respectively; the correlation coefficients in Oceania are relatively weaker, and the correlation coefficients are −0.664 and −0.611, respectively. In addition, we find that the correlation of import and export cargo flow in some regions is quite different: for example, the correlation of export flows in West Asia is weaker than that of import flows, with correlation coefficients of −0.432 and −0.707, respectively; the correlation of export flows in North Africa is significantly greater than that of import flows, with correlation coefficients of −0.603 and

−0.357, respectively. From the results, it can be found that the oil tanker trade in the crude oil export areas is mostly large-scale tankers. With the influence of economic policy and other factors, there is no large growth rate of the shipping situation of ports in these areas after the oil price slumped. At the same time, the situation of crude oil trade in the areas with more transit ports and import ports has a closer relationship with the change of oil price.

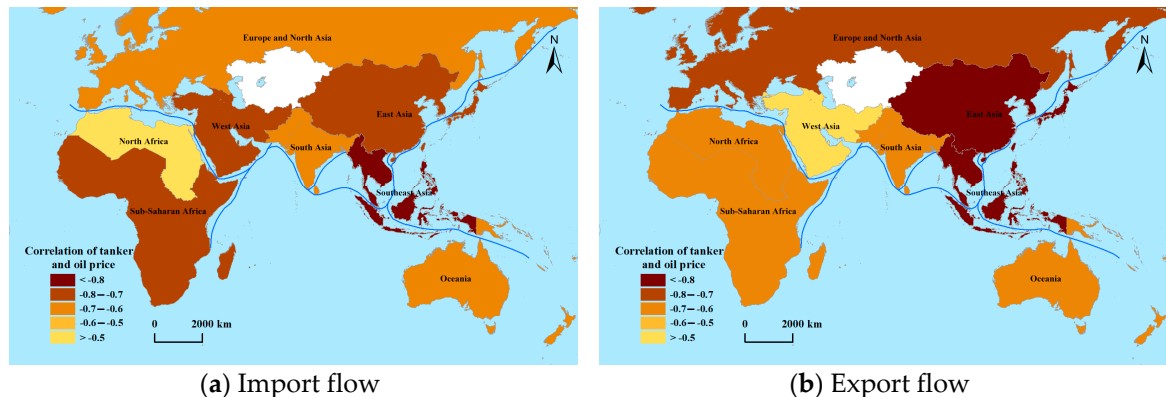

(**a**) Import flow        (**b**) Export flow

**Figure 6.** Correlation of tanker import and export.

### 4.2. Lag Analysis

From Figure 3, it can be seen that the change of oil tanker situation has obvious time lag compared with oil price. In order to further explore the relationship between the MSR's overall tanker shipping situation and oil price, we adopt the ARDL model, and use months as the time scale to perform the model fitting calculation with the first-order, second-order, and third-order lag periods, respectively. The results are shown in Figure 7: Where (a), (b), and (c) corresponds to the fitting results of the first-, second-, and third-order ARDL models, respectively. The blue line represents the true values of the monthly ship frequency, the red line indicates the fitting values of the monthly ship frequency, and the purple line indicates the residual values between true values and fitting values.

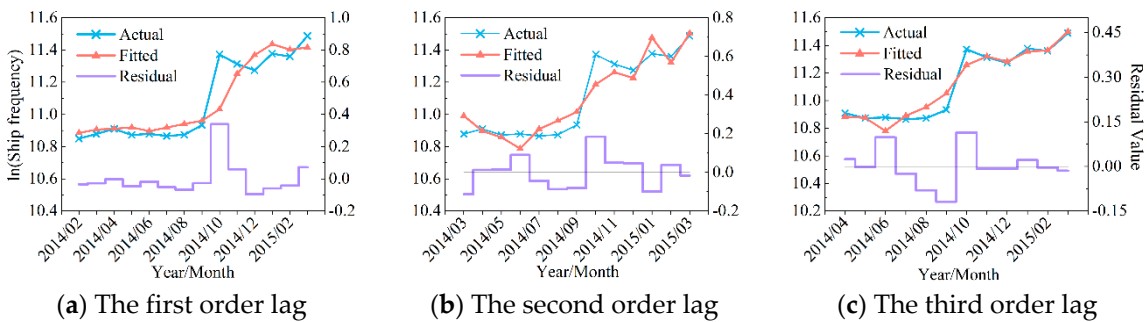

(**a**) The first order lag     (**b**) The second order lag     (**c**) The third order lag

**Figure 7.** The fitting results of tanker ARDL model.

Table 2 shows the corresponding results of the tanker fitting index calculations using different lag periods. According to the criteria of AIC and SC, the optimal delay period of the model is first determined as the third-order lag. The explanatory variables include the first-order, second-order, third-order lag term of ship frequency, as well as the current term of oil price, the first-order, second-order and third-order lag term of oil price. Then the variables with less reliability of fitting coefficients are eliminated by the *t*-test, that is, the third-order lag term of oil price $\ln P_{t-3}$ and the first-order lag item $\ln Q_{t-1}$. The adjusted tanker ARDL model makes all the explanatory variables highly significant. It can be seen from Table 2 that the adjusted model goodness-of-fit, Adjusted-$R^2$, is 0.871; the fitting effect of

tanker has not decreased. Finally, the expression of the ARDL lag relation of the whole MSR tanker is as follows:

$$\ln Q_t = 39.815 - 2.022 * \ln P_t + 0.908 * \ln P_{t-1} - 0.778 * \ln P_{t-2}$$
$$-1.232 * \ln Q_{t-2} - 0.636 * \ln Q_{t-2} \tag{4}$$

**Table 2.** The result of the tanker fitting index calculation.

| Lag Period | Adjusted-$R^2$ | F Statistical Quantity | AIC | SC |
|:---:|:---:|:---:|:---:|:---:|
| 1 | 0.762521 | 14.91392 | −1.122056 | −0.939469 |
| 2 | 0.822817 | 12.14529 | −1.342967 | −1.088221 |
| 3 | 0.824597 | 8.387541 | −1.412639 | −1.089367 |
| 3 * | 0.870976 | 15.85104 | −1.647591 | −1.405138 |

Note: * indicates the result of eliminating non-significant variables based on *T*-test.

From the fitting results of ARDL model of oil tanker, it can be seen that the months with large residual are concentrated in June–October. Oil price began to fall in June 2014, and tankers gathered to rise in October, reaching a sudden peak after a sharp fall in oil price. Then, the impact of the continuous decline in oil price on the shipping situation gradually diminished over time. Oil price have rebounded since December, and tanker shipments have gradually stabilized. The fitting results of ARDL model of tanker verify that the oil price slump has a significant time lag on the impact of oil tanker shipping situation, and we found that the optimal lag period for the change of tanker situation affected by the slumping oil price is three months. At the same time, the short-term impact of oil price has a certain timeliness.

*4.3. Regional Comparison and Analysis Along the MSR*

4.3.1. Lag Affects Regional Differences

According to the ARDL model fitting results of the whole tanker along the MSR, we bring the tanker frequencies in each study area into Equation (4) and calculate the residual of the monthly fitting value and statistical value of the model. The average (AVG) and mean square error (MSE) results of the residual values in each area are shown in Figure 8. The horizontal axis represents the areas along MSR: EA (East Asia), E&NA (Europe and North Asia), ESA (Southeast Asia), WA (West Asia), SSA (Sub-Saharan Africa), OA (Oceania), NA (North Africa), and SA (South Asia). The residual distribution along MSR is staggered, and the absolute values of AVG in North Africa and South Asia is greater than 0.5. The absolute values of AVG in other regions is less than 0.2. From the AVG, there is a big gap between the change of tanker shipping situation in each region and that of MSR as a whole. Among them, the AVG of East Asia, Southeast Asia, South Asia, and Oceania is above 0. Compared with the overall change of oil tanker shipping situation along the MSR, the increase of oil tanker shipping situation in these areas is higher than the overall average level. The AVG of Europe and North Asia, West Asia, sub-Saharan Africa, and North Africa is less than 0, and the increase of oil tanker shipping situation is lower than the overall average level along the MSR. In addition, the maritime shipping situation in North Africa and South Asia varies more significantly. In terms of the MSE, the distribution of residual in Southeast Asia, West Asia, North Africa, and South Asia fluctuates greatly, and the influence of oil price lag on the tanker shipping situation is relatively weak. The error distribution in Europe and North Asia, East Asia, Sub-Saharan Africa, and Oceania fluctuates slightly, and the influence of oil price lag on the tanker shipping situation is relatively strong. As Figure 9 shows, from the point of view of spatial distribution, the influence of oil price lag on both sides of the north and south is more obvious. The surrounding areas of the Malacca Strait, the North Indian Ocean, the Hormuz Strait, the Suez Canal, the Persian Gulf, and other important channels have a weaker lag impact on their shipping situation due to the slumping oil price. In regions with more oil importing

countries, such as Europe, East Asia, and East Africa, the lag of oil price has a stronger impact on the tanker shipping situation.

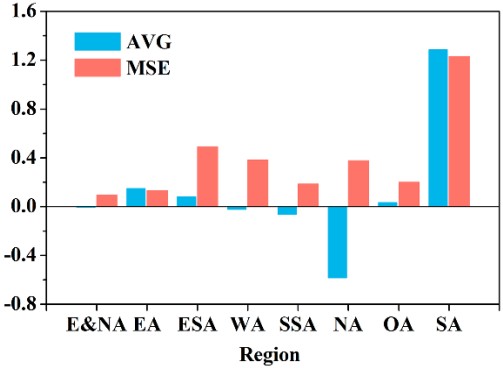

**Figure 8.** Residual calculation result.

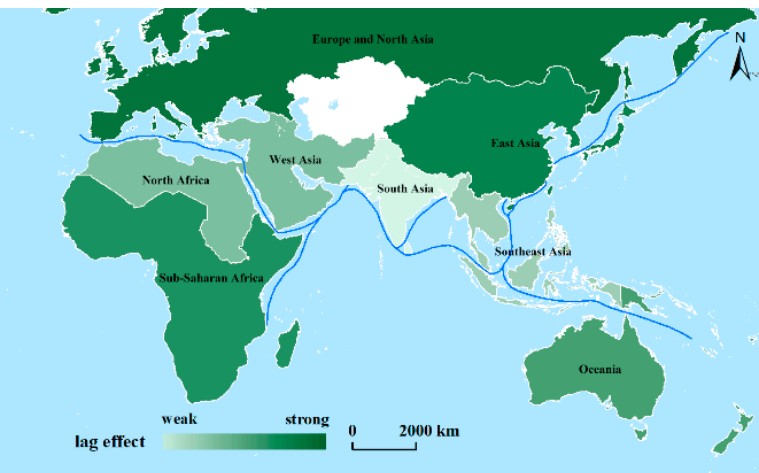

**Figure 9.** Regional distribution of lag effect.

### 4.3.2. Comparison of Cargo Flow before and after Oil Price Slumped

In order to further understand the impact of the oil price slump on the MSR tanker shipping situation, we selected the time intervals (June and October) before and after the oil price slumped based on the optimal lag period we found and obtained the structure map of the MSR crude oil transport flow, as shown in Figure 10. Nodes represent ports, and the size of nodes represents the frequency of trade within a month; the connection lines between nodes represent the trade between two ports, and the line width represents the frequency of trade on that route. The outermost label represents the port index number of the corresponding port in the WORLD PORT INDEX, for example, ZHOUSHAN port (59960). It can be found at the NGA Maritime Domain website (https://msi.nga.mil/NGAPortal/MSI.portal).

This paper chooses three typical regions from the perspective of crude oil import, export and transit. First, as the gathering place of global oil consumption and import, the ports in East Asia mainly connect the crude oil exporting countries and the transit hub ports of Malacca Strait, such as ZHOUSHAN port in China, CHIBA KO port in Japan, and ULSAN port in Korea. As Figure 10a,b shows, tanker cargo flows in East Asia are mainly concentrated within the region: between ports along Japan, South Korea and East China. In October after the oil price slumped, the frequency of port trade in these areas reached its peak, the frequency of maritime trade in CHIBA KO port increased from 1506 to 2405, and the frequency of ZHOUSHAN port increased from 876 to 106. Even the trade frequency of ports such as KAWASAKI KO port and QINGDAO port increased exponentially. At the same time, the hub ports of these crude oil importing countries have increased significantly with the maritime

trade of JURONG port and KEPPEL port in Singapore. Of the crude oil trade routes with ports in East Asia in June, 46 routes have had more than 30 trade frequencies, with the highest being ZHOUSHAN to ZHENHAI port (120 times); in October, after the oil price slumped, routes with frequencies over 30 had risen to 112, with the highest frequency being YOSU port to GWANGYANG port (410 times).

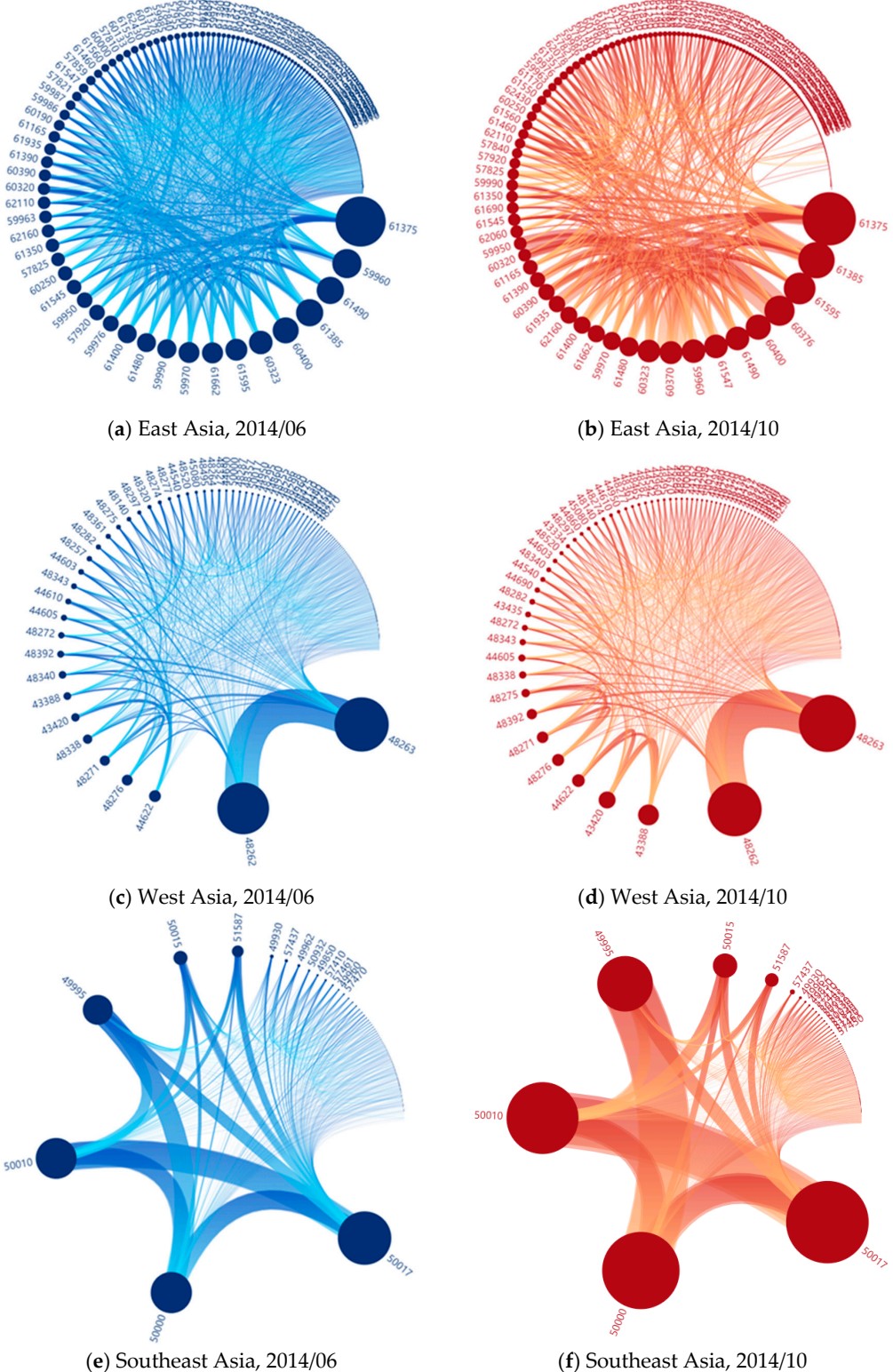

**Figure 10.** Crude oil transport flow distribution before and after oil price slumped.

Second, West Asia is the region with the richest oil reserves, with the largest production and the largest export volume in the world, mainly including Saudi Arabia, the United Arab Emirates, Iraq and other oil exporters. From Figure 10c,d, it can be seen that most ports in West Asia trade crude oil directly with the hub ports of oil importing countries such as East Asia and Europe, and transshipment ports such as KEPPEL port and JURONG port in Singapore. Tanker cargo flows are mainly concentrated in the Persian Gulf and around the Mediterranean Sea. After the oil price slumped, port trade in West Asia has increased slightly, but the increase is far from that in East Asia. In June, the trade frequencies of KHAWR FAKKAN port and FUJAYRAH port in the United Arab Emirates were 1630 and 1555, respectively. By October, the trade frequencies of the two ports were still in the top ranking, with a slight increase to 1717 and 1625. In addition, the trade frequency of AMBARLI port and ISTANBUL port increased significantly. Before the crude oil price slumped, 34 crude oil trade routes in West Asia had more than 20 frequencies; the highest route was KHAWR FAKKAN port to FUJAYRAH port (481 times). After the oil price slumped, 31 routes had more than 20 frequencies; the highest route was still KHAWR FAKKAN port to FUJAYRAH port (544 times).

Thirdly, Southeast Asia, especially the ports around the Strait of Malacca, relies on its geographical advantages to become an important regional hub for the transfer and convergence of global maritime trade. From Figure 10e,f, it can be seen that the distribution of cargo flow of maritime trade in Southeast Asia presents an obvious hub-and-spoke structure, and its crude oil trade mainly concentrates on the ports of Singapore and Malaysia. The cargo flow of these hub ports is mainly directed at the ports of oil exporters in West Asia and oil importers in East Asia, Europe and Africa. After the oil price slumped, Singapore's port trade frequency increased sharply, JURONG port's trade frequency increased from 3800 to 8689, KEPPEL port's trade frequency increased from 2929 to 8147, and Thailand's MAP TA PHUT port and BANGKOK port's trade frequency also increased exponentially. Among the crude oil trade routes with Southeast Asian ports in June, 31 routes had more than 30 frequencies, with the highest routes being PULAU BUKOM port to JURONG port (594 times), while 39 routes had more than 30 frequencies in October, with the highest still being PULAU BUKOM port to JURONG port (1969 times). Although crude oil transport routes have increased as a whole, the frequency of crude oil transport routes through transshipment hubs is far greater than that of other ports. This phenomenon fully proves that crude oil transport is mainly carried out by transshipment hubs for offshore crude oil trade.

## 5. Discussion

Based on the AIS trajectory data, this paper analyzes the impact of oil price on tanker shipping situation in areas along the MSR from the view of space-time: the correlation between oil tanker shipping situation and oil price is significant in time, and the correlation of the MSR maritime export cargo flow situation is higher than that of import cargo flow situation. The slumping oil price directly stimulates changes in crude oil supply and demand, raising the tanker trade in importing, exporting and transshipment hubs. At the same time, it is found that there is a three-month lag in the response process of tanker shipping situation to the change of oil price. From the perspective of the international environment, economic globalization has become the trend of world economic and trade development. On the one hand, the rapid development of the global economy has increased the demand for international crude oil, and on the other hand has also driven the development of international shipping demand. Due to the special status and role of international shipping companies and ports in economic globalization, adapting to economic transportation needs, being able to respond quickly and effectively in emergencies, and ensuring the sustainable development of shipping has become an important demand for maritime transportation. Moreover, significant changes in crude oil prices have a serious impact on maritime transport and will also affect shipping efficiency. Therefore, a reasonable grasp of the lag of tanker transport can provide a prejudgment for the operation and management of the transshipment hub, so that reasonable arrangements can be made in advance and transportation congestion and lack of berths can be avoided. It can bring the maritime market back to

a new equilibrium. In the meanwhile, it can also provide decision-making basis for oil consuming countries to make strategic policies in oil reserves and fuel management plans, thus ensuring the sustainable development of maritime crude oil trade along the MSR [49].

From the research results, we find that in the region with rapid economic development and energy demand expanding year by year, the shipping situation is obviously affected by the lag of oil price. However, that effect is relatively weak in the port areas which occupy the important hubs and are connected with many maritime trade areas. For example, Southeast Asia has superior natural conditions, especially the ports around the Malacca Strait that have significant geographical location and port condition advantages. Taking KEPPEL port of Singapore as an example, as a regional hub port, it plays an important role in the transfer and connection of maritime crude oil trade. On the current pattern of maritime trade structure, the advantages of the special maritime network in Southeast Asia have greatly improved the anti-interference capability of the change of the shipping situation on oil price. At the same time, the port infrastructure and service level have met the supply, warehousing and other needs of passing vessels [50] to make sure the route is clear. West Asia and North Africa also have important shipping hubs (the Strait of Hormuz and Suez Canal), so the shipping situation in these regions is less affected by the lag of oil price. In addition, in recent years, economic activities, production and consumption is increasing day by day and the maritime trade is also increasing year by year in East Asia, Europe, South Africa, East Africa and other regions. Especially in East Asia, Japan, South Korea and other major oil-consuming countries, oil demand is large, so the shipping situation in the region is vulnerable to the impact of the oil price slump, and the impact of the lag is even more obvious. Moreover, ports in South Asia are constrained by their hinterland traffic and infrastructure conditions, and their maritime trade is declining and less affected by such impact.

Tanker transportation is mainly responsible for the global maritime trade in petroleum and refined oil. The maritime crude oil trade depends on the structure of supply and demand, so tanker transportation has a direct and close relationship with crude oil export and consumption. Port conditions in many oil importing countries do not meet the ultra large crude carrier (ULCC), so crude oil transportation requires transit hub ports and then is shipped to import destinations by small vessels [35]. Comparing the changes of oil tanker cargo flow before and after the oil price slumped, we find that the shipping situation of importing countries and transshipment hub ports are concentrated rapidly. For example, the share of tanker transportation in the East Asia region rise from 93.7% to 95.6%, and that in the Southeast Asian region increased from 88.5% to 93.9%. Therefore, transit port and hub port of oil importing country play an important role in maritime crude oil trade. In summary, we put forward the following suggestions: (1) enhancing the bearing capacity of leading transshipment ports in tanker transportation system, ensuring the efficiency and sustainability of port operation when the shipping situation gathers, and reducing the risk of traffic congestion along the MSR. (2) Invest in the construction of potential ports with prominent geographical advantages in importing countries, such as QINGDAO port and DALIAN port in China, and TAKAMATSU port and KOMATSUSHIMA port in Japan. The construction of facilities and conditions of potential ports should be strengthened to meet the berthing needs of large ships, so as to develop potential ports into new import hub ports, share the transportation pressure of existing leading hub, and improve the smoothness of tanker shipping. (3) For ports in regions with strong delays, it is necessary to monitor the fluctuation of oil price in time and predict the development trend of crude oil trade rationally and formulate a reasonable port response plan to avoid risks, such as cargo accumulation, ship congestion, and ship delay, so as to ensure the smooth operation of ports and the sustainability of crude oil trade.

Compared with previous studies on maritime crude oil trade, we find that there are obvious differences in the impact of oil price in different regions. Before and after the oil price slumped, we analyzed the changes of oil import, export and transshipment areas in the short term based on the port-scale oil tanker cargo flow. Of course, the change of oil price itself is a dynamic process of long-term fluctuations. Our research mainly focuses on the situation of oil price slump, in the short-term, without considering the relationship between the trend of tanker shipping and oil price

rebound and the long-term fluctuation after that. In addition, the ARDL model is essentially a linear relationship, so it will be a future research direction to study whether oil price and shipping have a non-linear causality. Fuel, as one of the shipping costs, has different grades of fuel and different types of cargo ships have different impacts on transport costs. Therefore, future research can also combine container ships, dry bulk carriers and different grades of fuel to explore the impact of oil price on the shipping situation.

## 6. Conclusions

This paper establishes the lag relationship between ship frequency and oil price by using the ARDL model and compares the differences between different regions along the MSR affected by the oil price slump and changes of tanker cargo flow distribution. The following conclusions are drawn: (1) There is a significant negative correlation between tanker shipping frequency and oil price in the same period, that is to say, the shipping frequency will increase correspondingly when oil price slumps. The correlation between maritime export cargo flow situation and oil price is stronger than that of import cargo flow situation. (2) The impact of slumping oil price on the overall maritime trade along the MSR is lagging, with the lag period of three months. After the oil price slumped, the shipping situation of crude oil import in East Asian rose substantially, and the lag affected by oil price was obvious. However, Southeast Asia, which occupies important transshipment hub ports, is less affected by the lag of oil price. (3) After the oil price slumped, tanker cargo flow increases slightly at the stage of oil export, but most obviously at the stage of post-transit in maritime trade, especially between transit hubs and importing countries, and between ports within importing countries. (4) The sustainable development of crude oil trade along the MSR can be promoted by strengthening the carrying capacity of transshipment hub ports, increasing the investment in the construction of import potential ports (Qingdao port, TAKAMATSU port, etc.), and optimizing the structure system of the MSR maritime crude oil trade network.

**Author Contributions:** All of the authors contributed to the work in the paper. Specifically, Conceptualization, N.M. and H.Z.; data curation, Y.X. and T.Y.; investigation, N.M. and H.Z.; methodology, N.M. and Y.X.; supervision, H.Z.; validation, Y.X.; visualization, Y.X.; writing—original draft, N.M. and Y.X.; writing—review & editing, N.M., Y.X. and Y.R.K.

**Funding:** This research was funded by the National Natural Science Foundation of China (Grant No.41771476, 41771436), Natural Science Foundation of Shandong Province (Grant No. ZR2016DM02), and the Key Project of the Chinese Academy of Sciences (Grant No. ZDRW-ZS-2017-4-3).

**Conflicts of Interest:** The authors declare no conflicts of interest.

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
