# Peer review of "The Impact of Slumping Oil Price on the Situation of Tanker Shipping along the Maritime Silk Road"

_sustainability, doi:10.3390/su11174796_

Round 1

Reviewer 1 Report

The manuscript is well prepared.  I have no special comment.

Author Response

Point 1: The manuscript is well prepared. I have no special comment.

Response 1: Thank you so much.

Reviewer 2 Report

The paper discusses about the impact of slumping of fuel oil prices on the situation of tanker shipping. Following are my comments pertaining to the paper,

 Despite the main contributions are clearly stated in the document, motivation is still poor. Moreover, main contributions appear as the realistic advances with respect to previous results but only the contributions attained by authors appear as a summary of some relevant results. 

Authors need to present and summarize the research gaps coming out of the literature review and moreover the research gaps need to be properly connected with the contribution to make it look relevant for addressing.

Moreover, provide the appropriate reasoning for performing spearman's rank correlation coefficient and present more description to highlight the insights obtained from the results.

It is important to understand the impact of sustainability on the slumping of oil prices and it need to be properly established in the paper given that the huge effect of sustainability on shipping network.

Moreover, there are different grades of fuel, which can influence the cost and emission. How the fuel oil switching is considered from the perspective of slumping of oil prices? 

Authors have missed out citing some of the relevant papers in the domain of shipping operations from top journals. Kindly related the following works in line with the dynamic fuel pricing scenario and also sustainability aspect.

Bunkering Policies for a Fuel Bunker Management Problem for Liner Shipping Networks, European Journal of Operational Research

Hybridizing basic variable neighborhood search with particle swarm optimization for solving sustainable ship routing and bunker management problem, IEEE Transactions on Intelligent Transportation Systems

A hybrid dynamic berth allocation planning problem with fuel costs considerations for container terminal port using chemical reaction optimization approach, Annals of Operations Research

Fuel bunker management strategies within sustainable container shipping operation considering disruption and recovery policies, IEEE Transactions on Engineering Management

It seems the manuscript is technically fairly well written, but the managerial implications and the conclusion are weak. It would be nice if the managerial implications and the conclusion are stronger and more practically insightful.

Present a justification to strengthen the reason for the choice of performing correlation and lag analysis. 

Based on the above comments, I suggest major revision for the paper.

Author Response

Point 1: Despite the main contributions are clearly stated in the document, motivation is still poor. Moreover, main contributions appear as the realistic advances with respect to previous results but only the contributions attained by authors appear as a summary of some relevant results.

Response 1: In the Section I Introduction, we emphasize the importance of the 21st Century Maritime Silk Road to the maritime crude oil trade. Compared with previous studies, this paper highlights the impact of slumping oil price on the tanker shipping situation in the short term, and explores the variation trend of maritime trade from the perspective of overall maritime transport, as well as import, export and transshipment.

Point 2: Authors need to present and summarize the research gaps coming out of the literature review and moreover the research gaps need to be properly connected with the contribution to make it look relevant for addressing.

Response 2: In the revised version, we add the literature review section. By comparing with previous studies, we highlight the contribution of this paper from the data and analysis entry point.

Point 3: Moreover, provide the appropriate reasoning for performing spearman's rank correlation coefficient and present more description to highlight the insights obtained from the results.

Response 3: The reasons for choosing the spearman's rank correlation method are explained in Section 3.2.1.

Point 4: It is important to understand the impact of sustainability on the slumping of oil prices and it need to be properly established in the paper given that the huge effect of sustainability on shipping network.

Response 4: Based on the recommended literature, we supplement the information in the significance and discussion section.

Point 5: Moreover, there are different grades of fuel, which can influence the cost and emission. How the fuel oil switching is considered from the perspective of slumping of oil prices?

Authors have missed out citing some of the relevant papers in the domain of shipping operations from top journals. Kindly related the following works in line with the dynamic fuel pricing scenario and also sustainability aspect.

Bunkering Policies for a Fuel Bunker Management Problem for Liner Shipping Networks, European Journal of Operational Research

Hybridizing basic variable neighborhood search with particle swarm optimization for solving sustainable ship routing and bunker management problem, IEEE Transactions on Intelligent Transportation Systems

A hybrid dynamic berth allocation planning problem with fuel costs considerations for container terminal port using chemical reaction optimization approach, Annals of Operations Research

Fuel bunker management strategies within sustainable container shipping operation considering disruption and recovery policies, IEEE Transactions on Engineering Management

Response 5: The proposal that we should consider the impact of different grades of fuel on shipping costs is significant. However, it is difficult for us to obtain this data at present. We will seriously consider this suggestion in future studies. By referring to the recommended literature, we explain this suggestion in the future research direction of the discussion section. And we have cited recommended literatures in the paper.

Point 6: It seems the manuscript is technically fairly well written, but the managerial implications and the conclusion are weak. It would be nice if the managerial implications and the conclusion are stronger and more practically insightful.

Response 6: In the introduction section, the significance of the research results to the port operation management and the sustainable development of maritime crude oil trade is supplemented. Besides, the suggestions and significance to the optimization of the MSR shipping network are added in the conclusion.

Point 7: Present a justification to strengthen the reason for the choice of performing correlation and lag analysis.

Response 7: In the analysis of the section 4, we analyze the experimental results in depth, and supplement the preconditions of using these experimental steps.

Reviewer 3 Report

This article provides a very interesting analysis of the crude oil trade and its relation with the crude oil price.

would it be possible to add figures with the evolution of the crude oil in the period analyzed would it be possible to add a map with the ports analyzed clearly named so we can have a better vision of them? I would appreciate if authors can explain the period more deeply, why from January 1st, 2014 till March 2015?  I also would appreciate if authors can explain the reason why they looked at these ports.

Author Response

Point 1: This article provides a very interesting analysis of the crude oil trade and its relation with the crude oil price.

Response 1: Thank you very much.

Point 2: would it be possible to add figures with the evolution of the crude oil in the period analyzed would it be possible to add a map with the ports analyzed clearly named so we can have a better vision of them? I would appreciate if authors can explain the period more deeply, why from January 1st, 2014 till March 2015?  I also would appreciate if authors can explain the reason why they looked at these ports.

Response 2: The reason for the time period and ports selection is supplemented in 3.1.

The port in the map is not clear enough because there are many ports involved in this paper, which cannot be labeled in a map. However, in order to enable readers to clearly understand the trade changes between ports that we focus on in the comparison of cargo flows, we changed Figure 10 into Chord diagrams.

Round 2

Reviewer 2 Report

Authors have adequately addressed my comments and now I feel the paper is at par with the standard of the journal and it can be accepted for publication in the journal.